

# Idiosyncratic liver pigment alterations of five frog species in response to contrasting land use patterns in the Brazilian Cerrado

Lilian Franco-Belussi[1,2], Diogo B. Provete[1,3], Rinneu E. Borges[4], Classius De Oliveira[2] and Lia Raquel S. Santos[5]

[1] Instituto de Biociências, Universidade Federal de Mato Grosso do Sul, Campo Grande, Mato Grosso do Sul, Brazil
[2] Departamento de Biologia, Universidade Estadual Paulista, São José do Rio Preto, São Paulo, Brazil
[3] Gothenburg Global Biodiversity Centre, Gothenburg, Västra Götaland, Sweden
[4] Departamento de Biologia, Universidade de Rio Verde, Rio Verde, Goias, Brazil
[5] Instituto Federal de Educação, Ciência e Tecnologia Goiano, Rio Verde, Goias, Brazil

Corresponding author
Lilian Franco-Belussi, lilian.belussi@gmail.com

## ABSTRACT

**Background:** Changes in land use trigger environmental changes that can lead to decreased biodiversity and species loss. The liver is an essential detoxification organ that reflects systemic physiological responses to environmental changes. Here, we tested whether contrasting land use patterns influence the amount of substances from the hepatic cellular catabolism and melanomacrophages (MMs) of five anuran species in the Brazilian Cerrado.

**Methods:** We collected the same five species of pond-dwelling frogs in one protected area and in an area with intense agricultural activity. We used routine histological and histochemical techniques to quantify the area occupied by lipofuscin, melanin, and hemosiderin in the liver of two frogs *Leptodactylus fuscus*, *Physalaemus cuvieri*, and three tree-frogs *Dendropsophus minutus*, *Scinax fuscomarginatus*, and *Boana albopunctata*. We classified land use types in a buffer around each pond based on satellite images. We then used a double-constrained Correspondence Analysis, a recently developed ecological method to relate functional traits to environmental variables, to test the effect of each land use type on the area of each liver pigment.

**Results:** There was an increase in the amount of melanin in environments with high proportion of agriculture, as well as variation in the amount of lipofuscin and hemosiderin. Liver pigments of *P. cuvieri* and *B. albopunctata* varied more strongly in response to land use types, suggesting they could be good indicator species. Therefore, the area of MMs in the liver and the metabolic products in their cytoplasm can be used as biomarkers of environmental changes in regions with intense agricultural activities. Our results add a new perspective to the influence of land use patterns on environmental health by highlighting the effect of environmental changes on internal morphological aspects of animals.

## INTRODUCTION

Human-driven land use changes are causing biodiversity loss around the world (*Newbold et al., 2015*; *Powers & Jetz, 2019*). Brazil is one of the countries with the highest proportion of net loss of tree cover in South America, with a loss of 8% from 1982 to 2016 (*Song et al., 2018*). At the same time, there was a 12% increase in short vegetation cover (*Song et al., 2018*), which includes shrubs and herbaceous vegetation. This trend was especially significant in the Brazilian Cerrado, of which less of 2% are protected (*Beuchle et al., 2015*; *Françoso et al., 2015*; *Damasco et al., 2018*). Accordingly, more than half of the original 2 million km$^2$ of the Cerrado were transformed into planted pastures and annual cultures by 2005 (*Klink & Machado, 2005*). Central Brazil is a thriving region for industrial agricultural activities (*Dias et al., 2016*) as one of the largest exporters of soybean and cattle meat in the world (*Contini & Martha, 2010*; *Reynolds et al., 2016*). One of the consequences of land use change for export-oriented agricultural activities is the intensive use of agrochemicals (*Schiesari et al., 2013*; *Aranha & Rocha, 2019*). Therefore, land use changes, along with agrochemicals, are currently the main threats for biodiversity conservation and the sustainable use of natural resources in this biome (*Reynolds et al., 2016*).

Water quality in the Cerrado has been drastically affected by the intense use of fertilizers, with a significant increase in nitrogen and pesticides (*Hunke et al., 2015*). As a result, freshwater habitats receive a great load of contaminants, which impact several aspects of aquatic biodiversity (*De Marco et al., 2013*; *Bichsel et al., 2016*). For example, there is evidence that the replacement of natural habitats by urban and agricultural areas decrease not only amphibian populations, but also their genetic diversity (*Eterovick et al., 2016*). Additionally, land use changes can promote rapid transformation in biological communities beyond species composition. Specifically, it can alter phenotypic aspects of several animal groups (*Borges et al., 2019a*), which impact how species interact and adapt to a changing environment. These phenotypic changes include DNA damages (*Borges et al., 2019b*) and developmental abnormalities (*Borges et al., 2019a*). However, little is known about the effects of contrasting land uses on internal phenotypic aspects of organisms inhabiting Neotropical agroecosystems.

Amphibians are good bioindicators of environmental quality because they show rapid responses to environmental stress (*Halliday, 2000*; *Brodeur & Vera Candioti, 2017*). In addition, they have permeable skin and eggs, making them vulnerable to aquatic contamination and infections. As such, these animals are useful for monitoring changes in both the aquatic and terrestrial environments because they depend on the two environments to complete their life cycle (*Brodeur & Vera Candioti, 2017*). Amphibians are rapidly declining worldwide due to human-induced changes in the environment (*Catenazzi, 2015*; *Alton & Franklin, 2017*). Several factors are involved in this decline, including climate change, increased incidence of ultraviolet (UV) radiation due to ozone depletion, invasive species, environmental contamination, diseases, and habitat change or loss (*Collins & Crump, 2009*; *Alton & Franklin, 2017*).

Previous studies have analyzed the effects of agrochemicals on tadpole developmental abnormalities (*Borges et al., 2019a*) or genotoxic effects in adult anurans. However, environmental alterations that promote morphological and physiological changes at the tissue level are poorly understood. The liver plays a key role in the detoxification of xenobiotics (*Pérez-Iglesias et al., 2018*; *Fanali et al., 2018*), as well as other functions related to the metabolism (*Hinton, Segner & Braunbeck, 2001*; *Fenoglio et al., 2005*). The detoxification is performed by hepatocytes and melanomacrophages (MMs) and may be either enzymatic or not. Melanomacrophage centers and their pigments (melanin, lipofuscin, and hemosiderin) are involved in the hepatic response to various toxic compounds. Thus, the liver is an important organ to evaluate the response of organisms to environmental pollutants (*Fenoglio et al., 2005*).

Melanomacrophages are cells that occur in the hepatic tissue and produce and store melanin in their cytoplasm. The area and occurrence of MMs in the liver increase in response to environmental stressors, such as temperature (*Santos et al., 2014*), UV radiation (*Franco-Belussi, Sköld & De Oliveira, 2016*), and xenobiotics (*Çakıcı, 2015*; *Pérez-Iglesias et al., 2016*). In addition to melanin, MMs contain catabolic substances originated from the degradation of red blood cells: hemosiderin and lipofuscin, originated from the degradation of polyunsaturated membrane lipids (*De Oliveira & Franco-Belussi, 2012*). Therefore, the density of MMs can be a useful morphological biomarker for the effect of environmental stressors (*De Oliveira et al., 2017*). Morphofunctional changes in the hepatic parenchyma happen as a result of contaminants, suggesting the action of detoxification by MMs due to the processing action of some enzymes, besides the protective action of melanin (*Fenoglio et al., 2005*). Additionally, the functions of MMs are also related to the maintenance of homeostasis, regulating fibrosis, controlling basophiles, and participating in the recycling of red blood cells (*Gutierre et al., 2018*).

Changes in the amount of catabolic substances within MMs may be associated with changes in phagocytic activity (*Bucke, Vethaak & Lang, 1992*; *Fenoglio et al., 2005*). For example, lipofuscin is produced as a result of the degradation of cellular components. Thus, its increase may hinder cell renewal and promote an accumulation of damaged cellular components in the tissue. The accumulation of lipofuscin can increase cellular sensitiveness to oxidative stress, since this molecule binds to metals, such as iron and copper (*Terman & Brunk, 2004*). Hemosiderin is an intermediate metabolite generated during the recycling of components in blood production. Thus, the accumulation of hemosiderin indicates a disorder in blood cell recycling (*Pérez-Iglesias et al., 2016*). In addition, environmental factors that alter the concentration of these substances in the liver may strongly contribute to the decrease in animal health. Consequently, this can affect individual fitness and promote population decline in the long term. Therefore, analyzing liver cell physiology may be useful for detecting the effects of environmental changes by human actions. However, while previous studies (*Franco-Belussi, De Lauro Castrucci & De Oliveira, 2013*; *Santos et al., 2014*; *Gregorio et al., 2016*; *Pérez-Iglesias et al., 2016*) have analyzed the variation of melanin, lipofuscin, and hemosiderin in response to climatic factors and xenobiotics in laboratory experiments using model species, no study
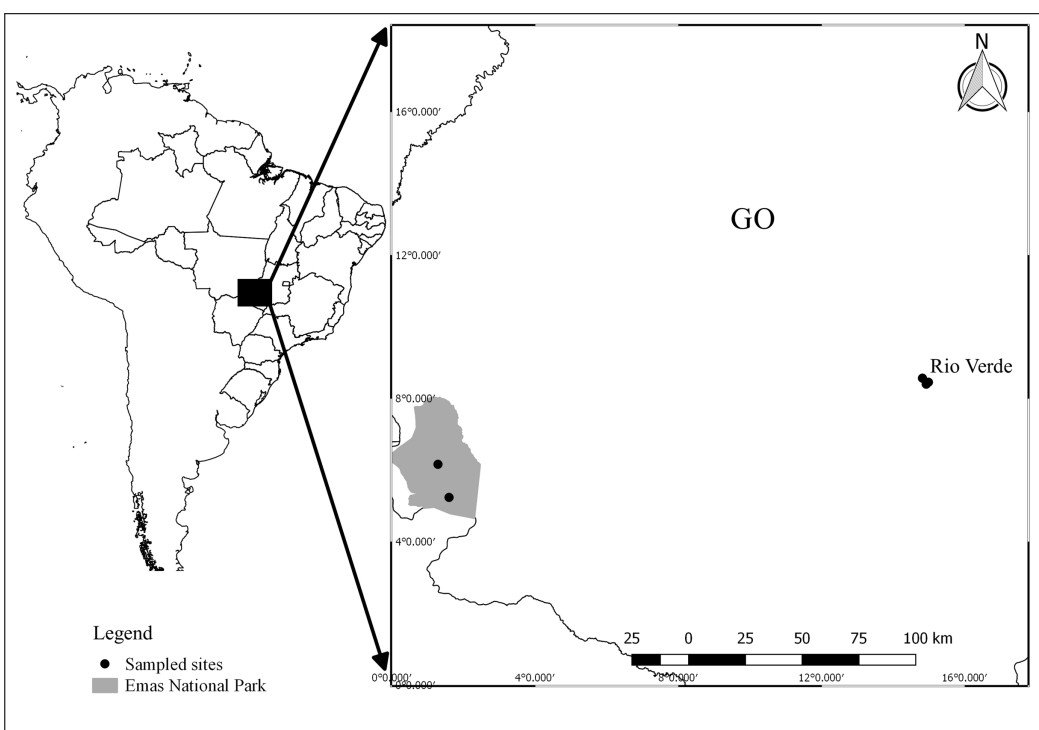

**Figure 1 Map showing the location of sampling sites in Goiás, Central Brazil.** Points represent ponds sampled. Image credit: R. F. Oliveira. 

has investigated how these three substances vary at the same time in response to contrasting land use patterns in wild amphibian populations.

Here, we tested the influence of contrasting land use regimes associated with aquatic contaminants on liver cell morphology and physiology of five frog species in the Brazilian Cerrado. These can be hidden effects of changes associated with agricultural intensification that are often neglected in biodiversity assessments that only consider species abundance and occurrence. We expect that frogs collected in ponds embedded in areas with intense agricultural activity will have increased hepatic metabolism because of potential aquatic contaminants, increased solar incidence and temperature. As a result, individuals will have higher amounts of melanin, due to its protective effects against free radicals in tissues, along with higher amounts of hemosiderin and lipofuscin, because these two substances reflect changes in hepatic metabolism.

## MATERIALS AND METHODS

### Study area and specimen sampling

Field work was carried out in two regions: three ponds in the surroundings of Rio Verde (17°48′11.28″ S; 50°56′24.95″ W), and two ponds within the Emas National Park (18°15′32.11″ S; 52°53′14.13″ W; Fig. 1), Goiás, central Brazil. The set of ponds were selected in both regions because they had the same species composition, allowing us to compare the effects of contrasting land use in a paired design (our dataset is available at *Franco-Belussi et al. (2020)*). Ponds in the National Park were selected based on a previous
survey (*Kopp, Signorelli & Bastos, 2010*) that sampled the same ponds and recorded the presence of our target species. Sampling sites in the Rio Verde region were selected because they were inserted into an agricultural matrix and also harbored the same species. Ponds were separated by at least 1.25 km and a maximum of 3.20 km in Rio Verde and by 15.30 km in the National Park. Samplings were conducted during the breeding season between November 2013 and March 2014. Field work consisted of 1 week of sampling in each region each year.

The region around the city of Rio Verde is a thriving agricultural area, mainly covered with planted pastures, and monocultures of soybean, corn, and sorghum. The Emas National Park is an enclave of Cerrado Protected Area with several typical vegetation types of this biome, varying from open formation, such as *campo limpo* and *campo rupestre*, to closed-canopy formations, such as *Cerradão* and Seasonal Deciduous Forest (*Valente, 2006*). The Instituto Chico Mendes de Conservação da Biodiversidade provided colleting permit (Sisbio #34485-1) and Emas National Park provided housing and authorization for field work.

We used a paired design in which we collected five adult males of the following species in each region: *Boana albopunctata, Dendropsophus minutus, Scinax fuscomarginatus, Leptodactylus fuscus*, and *Physalaemus cuvieri*. These species were previously selected because they occurred in both regions (*Kopp, Signorelli & Bastos, 2010*). Additionally, all of them are widely distributed throughout South America (*AmphibiaWeb, 2020*) and seem to be generalists, occurring in a wide range of habitats, from open, natural formations, to bush Savannas, to peri-urban areas (*Haddad et al., 2013*). All species are classified as Least Concern in both the IUCN Red list (*IUCN, 2010*) and Brazilian National Red list (*ICMBio, 2018*). *B. albopunctata, S. fuscomarginatus*, and *D. minutus* are hylid tree-frogs that call perched on the vegetation, while *L. fuscus* and *P. cuvieri* are medium-sized leptodactylids that are found calling and usually foraging on the ground near temporary ponds (*Haddad et al., 2013*). Both *B. albopunctata* and *D. minutus* deposit egg masses directly on water, while *L. fuscus* and *P. cuvieri* build foam nests in which they deposit eggs on the margin (in the case of *P. cuvieri*) or on subterranean chambers (in the case of *L. fuscus*; *Haddad et al., 2013*) of water bodies. Thus, adults, eggs, and larvae of these species have different degrees of contact with contaminated water. Consequently, these species can potentially be useful as environmental indicators, since they are highly abundant within their geographic range and seem to respond quickly to environmental disturbance.

## Water quality analysis

To test if the ponds used by amphibians were contaminated by pesticides, we collected one-L samples of water in one pond from each region. We took samples at approximately 10 cm depth from all ponds surveyed, but they were further grouped by region before analysis. We quantified organochlorines and organophosphates in only one water sample of each region. Samples were taken using a sterile, amber glass vial and immediately stored in ice at 4 °C, then sent to the laboratory and analyzed within 24 h. Quantification was made using standard methodology by A3Q Laboratory of Environmental Quality (Cascavel, Paraná, Brazil). Samplings from the Park did not contain any substance above

the references, while Rio Verde had atrazine well above the level allowed by the Brazilian environmental agency (Table S1).

## Routine histological processing

Specimens were anesthetized with five g/L benzocaine. This procedure was approved by the ethics committee on animal use of our university (protocol #0316, CEUA/UniRV). Liver fragments were extracted and fixed in metacarn solution for 3 h. Subsequently, they were dehydrated in an alcoholic series and included in historesin (Leica®, St. Gallen, Switzerland). We obtained two µm sections in rotating microtome (RM 2265; Leica, St. Gallen, Switzerland), which were stained with hematoxylin and eosin.

Histochemical analyzes were performed for the detection of lipofuscin as follows: sections were incubated for 15 min in Schmorl's solution, composed of 75 mL of 1% ferric chloride, 10 mL of potassium ferricyanide, and 15 mL of distilled water. Then, sections were immersed in neutral red aqueous solution at 1% followed by 1% aqueous solution of eosin. For the detection of hemosiderin, sections were incubated for 15 min in acid solution of ferrocyanide obtained by the dissolution of 2 g of potassium ferrocyanide in 100 mL of 0.75 mol/L hydrochloric acid solution. Subsequently they were immersed in aqueous solution of 1% neutral red followed by aqueous solution of 1% eosin.

Images were captured in a digital camera coupled with a microscope (Lab50AB-S; Laborana, São Paulo, Brazil) using an image capture system (Lab3001-C) and analyzed in Image Pro-Plus v.6.0 (Media-Cybernetics Inc., Rockville, MD, USA). Image analysis was conducted using 25 randomized histological fields for each animal. Specifically, we quantified the area occupied by each substance by differences in staining intensity in 25 pictures per animal following *Santos et al. (2014)*. Quantifications were done in a double-blind fashion.

## Quantification of land use pattern

To quantify land use, we used a shape file with land cover and use classes (*IBGE, 2018*; http://www.sieg.go.gov.br/produtosIMB.asp?cod=4725). This file classifies land use and cover into 14 classes based on satellite images from 2014 (see Appendix II in *IBGE (2018)*). To calculate the area of each land use, we established a buffer of 500 m radius around each water body sampled and calculated the area occupied by each land use class in ArcGIS 9.0 software (*Environmental Systems Research Institute (ESRI), 2011*). This buffer size has been commonly used in studies of landscape ecology involving anurans (*Almeida-Gomes, Rocha & Vieira, 2016*), since it is usually the dispersal distance of individuals moving among ponds in agricultural landscapes. The areas of each land use class were then converted into proportions and used as predictor variables in further analyzes. The land use types we found around our sampling sites were: natural grassland and shrub vegetation, artificial area, forest mosaic, grassland mosaic, planted pastures, and natural pasture, and agricultural area. Since the area of some land use types in our data set was small, we combined artificial area with grassland mosaic into a class called anthropized area, and also combined forest mosaic, natural pastures, and agricultural area into a class called farming to improve data analysis and interpretation of results.

## Data analysis

There are currently several methods to test the influence of environmental variables on species traits (see review in *Kleyer et al. (2012)*), the so-called fourth-corner problem. However, there is still no consensus as to which method is best, and it seems that this decision is dependent on the context and trait data type (continuous, categorical or mixed), although studies show that correlating Community-weighted Mean with environmental variables always seems to produce larger Type I Errors (*Peres-Neto, Dray & Ter Braak, 2017*; *Ter Braak, Peres-Neto & Dray, 2017*). Here, we used a double-constrained Correspondence Analysis (or dc-CA, for short) to test whether the mean area of melanin, hemosiderin, and lipofuscin (response variables) of the five species are correlated with different classes of land use (predictor variables). This is a method proposed long ago (*Lebreton et al., 1988*), but a new algorithm has only recently been proposed (*Ter Braak, Šmilauer & Dray, 2018*). The advantages of dc-CA are that it considers the correlation among environmental variables and among traits, when relating traits to environment (*Ter Braak, Šmilauer & Dray, 2018*), differently from RLQ and CWM-RDA (*Kleyer et al., 2012*).

dc-CA uses three tables: a species-by-site matrix $\mathbf{Y}$, which can contain either abundances or incidence; a site-by-environment matrix $\mathbf{E}$, and a species-by-trait $\mathbf{T}$ matrix. The method starts by conducting a Correspondence Analysis constraining both the columns (species) by species traits and row scores of the $\mathbf{Y}$ matrix by environmental variables (*Ter Braak, Šmilauer & Dray, 2018*). We then produced a triplot showing the relationship among traits, environmental variables, and species incidence in a single ordination diagram. Analysis was conducted using R code (*R Core Team, 2020*) available in the Supplemental Material of *Ter Braak, Šmilauer & Dray (2018)* and Canoco 5.12 (*Ter Braak & Šmilauer, 2018*). All data and an R Markdown dynamics document with code used to anlayze the data is available at *Franco-Belussi et al. (2020)*.

# RESULTS

The amount of each pigment varies among species (Fig. S1A), but *P. cuvieri* and *B. albopunctata* had the highest areas of melanin, while *D. minutus* and *L. fuscus* had relatively more lipofuscin. We found both great intra- and interspecific variation in the amount of each substance (Fig. 2). Most species had higher amounts of lipofuscin and melanin than hemosiderin, except *B. albopunctata* that had lower amounts of lipofuscin than the other species. Interestingly, *L. fuscus* and *D. minutus* showed little intraspecific variation in the amount of all three substances, while *P. cuvieri* and *B. albopunctata* had high intraspecific variation. The means for the three substances was similar for *L. fuscus* and *D. minutus*, whereas the other three species had very distinct means. Interestingly, the relative proportion of the three pigments also varied between sampling regions, because animals from the agricultural area had more melanin and lipofuscin, but less hemosiderin (Fig. S1B). Species whose liver pigments more strongly varied between sampling regions were *P. cuvieri*, *B. albopunctata*, and *S. fuscomarginatus* (Fig. 3).

The total inertia of the dc-CA model was 0.955. The first axis of dc-CA accounted for 72% of the variation in the trait-environment relationship, while the second accounted for 26%. The maximized fourth-corner correlation along the first and second axes are

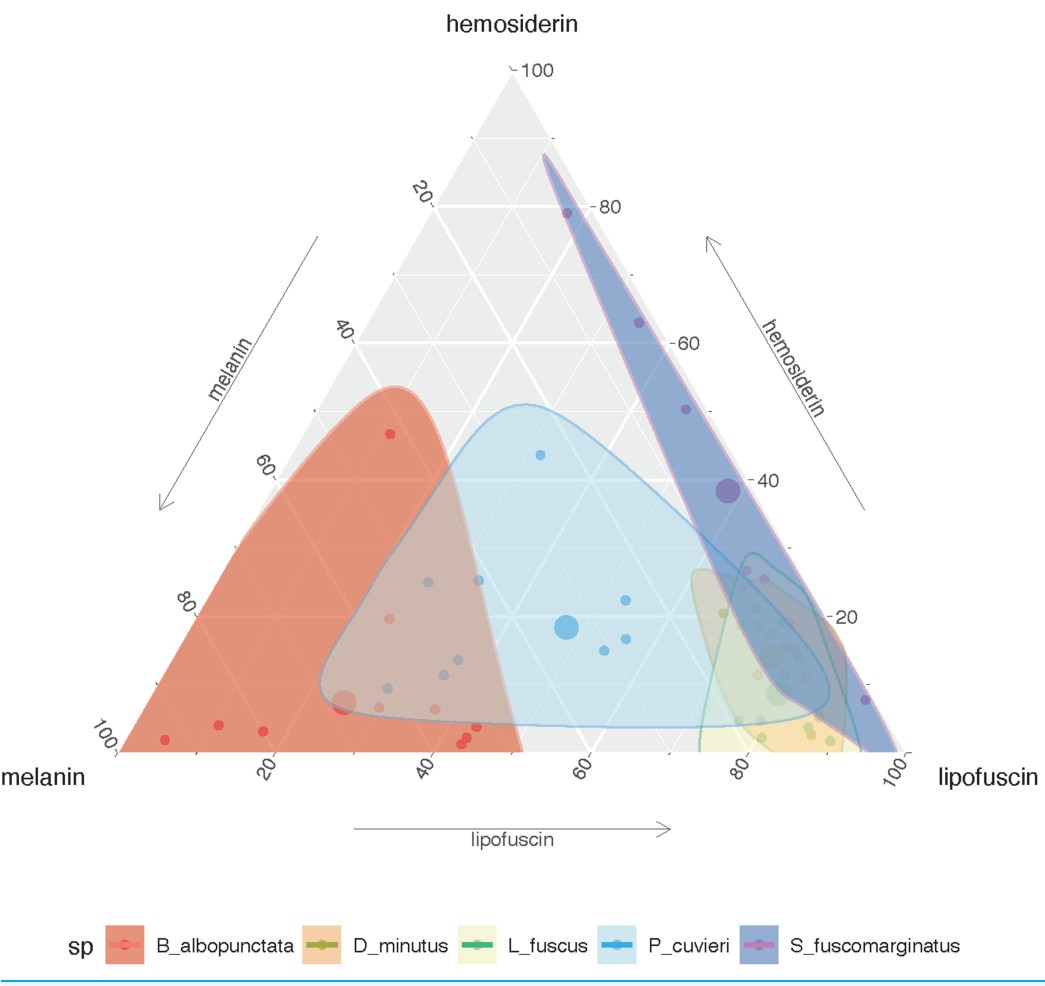

**Figure 2 Ternary plot showing the relative proportion of the area of each substance in the five frog species considering all sampling sites.** Small points represent individual measurements, while the large dot represents the mean of each substance for each species.

0.83 and 0.50, respectively. Melanin was positively correlated with natural and planted pastures, but negatively correlated with man-made buildings and agriculture (Table 1). Hemosiderin was negatively correlated natural and planted pastures, and man-made buildings, but positively correlated with agriculture. The correlation pattern of lipofuscin was almost identical to hemosiderin with small changes in the strength of correlation with some land use types (Table 1).

The abundance of *P. cuvieri* was positively correlated with the percentage of planted pasture in the landscape (Fig. 4). This was also the species with the highest amount of melanin in the liver (Fig. 4), which was positively correlated (fourth-corner correlation = 0.473, Table 1) with the percentage of planted pasture (Fig. 4). In contrast, *L. fuscus* had the lowest amount of melanin (Fig. 4) whose abundance was negatively correlated with the percentage of planted pasture in the landscape (Fig. 4).

Conversely, the abundance of *D. minutus* and *S. fuscomarginatus* were positively correlated with area of agriculture and negatively with anthropogenic area, while the abundance of *B. albopunctata* was positively correlated with anthropogenic area and

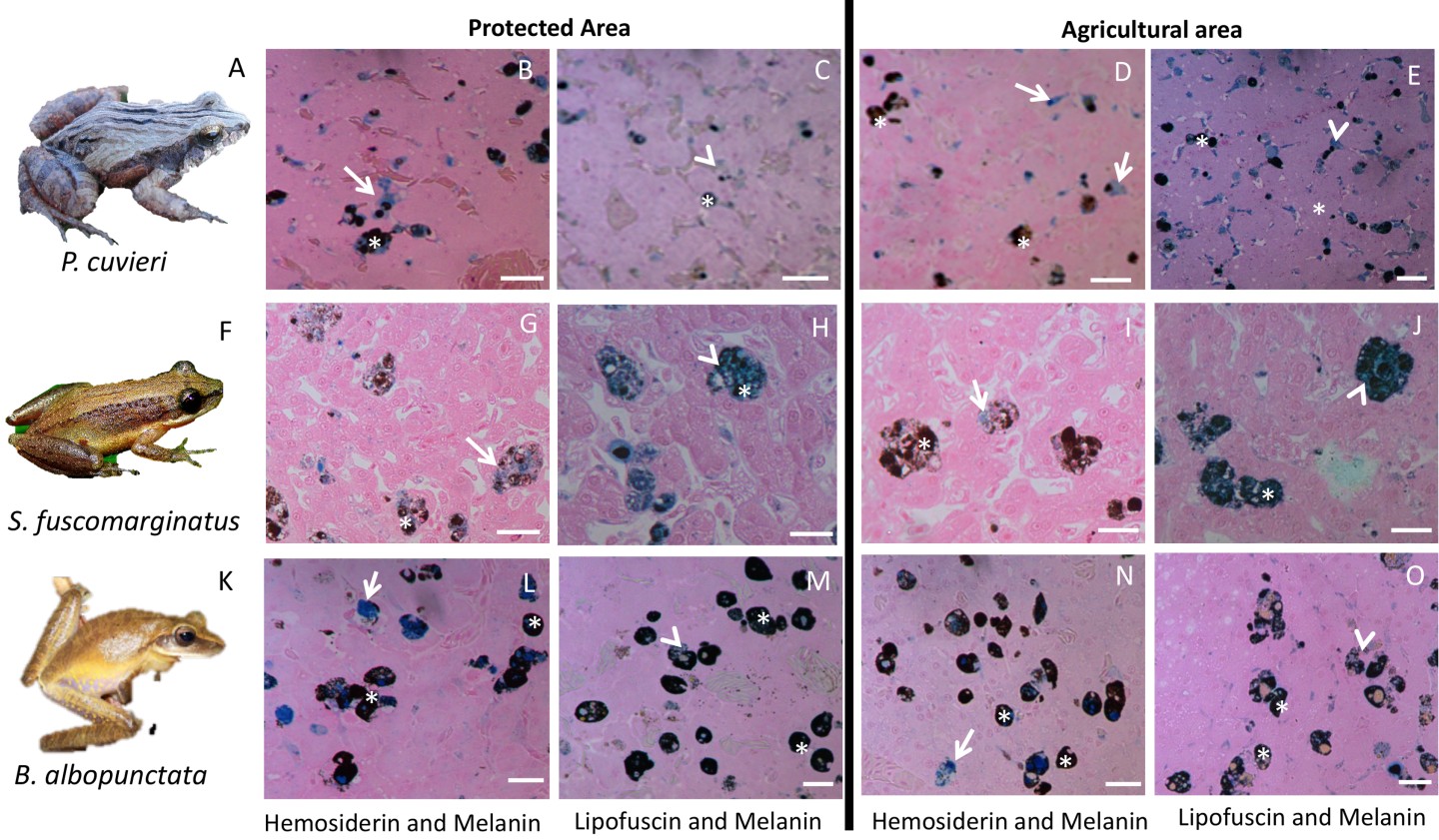

**Protected Area** | **Agricultural area**

Hemosiderin and Melanin | Lipofuscin and Melanin | Hemosiderin and Melanin | Lipofuscin and Melanin

**Figure 3** **Plate with histological sections showing differences in liver pigments for the three species that had the highest change in mean pigment area between the two sampling regions.** (A–E) *P. cuvieri*. (F–J) *S. fuscomarginatus*. (K–O) *B. albopunctata*. Dark blue color corresponds to hemosiderin, while grayish blue color corresponds to lipofuscin. Legend: arrow = hemosiderin, arrowhead = lipofuscin; asterisk = melanin. Scale bar = 25 µm. Staining: Schmorl's solution for lipofuscin and acid solution of ferrocyanide for hemosiderin. *S. fuscomarginatus* (F) photo credit: Mario A. Sacramento.                

negatively with area of agriculture (Fig. 4). *D. minutus* and *S. fuscomarginatus* had higher amounts of hemosiderin and lipofuscin, suggesting higher hepatic metabolism, which was positively correlated with agriculture and negatively with percentage of anthropogenic area (Fig. 4). Conversely, *B. albopunctata* had fewer hepatic catabolite substances (Fig. 4).

## DISCUSSION

We found that *P. cuvieri* occurred in sites with planted pastures and man-made buildings, had higher amounts of melanin, while the abundance of *L. fuscus* was positively correlated with man-made buildings and negatively correlated with planted pastures. Additionally, *D. minutus* and *S. fuscomarginatus* had high amount, while *B. albopunctata* had low amount of lipofuscin and hemosiderin and its abundance was positively correlated with man-made buildings. Taken together, these results demonstrate the effects of environmental changes on MMs and hepatic metabolism of frog species.

The amount of melanin was highest in *P. cuvieri* and *B. albopunctata*, but low in *L. fuscus*, *D. minutus*, and *S. fuscomarginatus*. Despite this interspecific variation, there was a clear pattern of change in melanin within species between sampling regions. There was

**Table 1 Fourth-corner correlation calculated between species traits (area of each substance in the liver) and environmental variables (percentage of land use type in a 500 m buffer around ponds).**

|  | Melanin | Hemosiderin | Lipofuscin |
|---|---|---|---|
| Natural pasture | 0.139 | −0.276 | −0.252 |
| Man-made building area | −0.033 | −0.188 | −0.228 |
| Planted pastures | 0.473 | −0.135 | −0.267 |
| Agriculture | −0.386 | 0.721 | 0.78 |

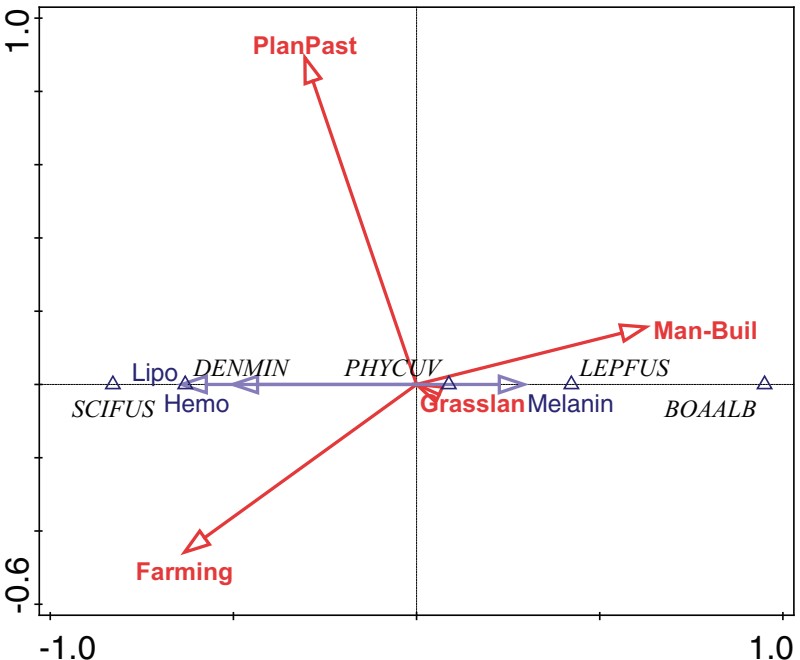

**Figure 4 Ordination diagram of double-constrained correspondence analysis.** The relationships among species abundance, liver cell metabolic pigments, and land use classes are shown. Arrow length indicates the importance of variables for the construction of ordination axes.

also a high positive correlation between melanin and planted pastures (0.473), and a less strong correlation with man-made buildings (−0.033), and natural pastures (0.139). Melanin is a complex polymer that absorbs and neutralizes free radicals, besides participating in the innate immune response and protection of tissues in ectotherms (*Cesarini, 1996*). Changes in the amount of melanin was found in frogs experimentally exposed to several xenobiotics. In these experiments, the variation in melanin was due to several compounds (reviewed in *De Oliveira et al. (2017)*), such as polycyclic aromatic hydrocarbons (PAHs; *Fanali et al., 2018*), herbicides (e.g., atrazine and glyphosate; *Pérez-Iglesias et al., 2019*; *Bach et al., 2018*), drugs, and endocrine disrupters (*Gregorio et al., 2016*). Here, we found large amounts of atrazine in the area under agriculture influence (more than 5,000 times the limit accepted by Brazilian legislation, i.e., 5,349.940 μg/L). Atrazine is an endocrine disruptor, which has immunotoxic and immunosuppressive

effects even at concentrations usually found in the environment (i.e., <500 µg/L; *Rohr & McCoy, 2010*). This substance can change swimming ability, cause malformations, and promote nuclear alterations at higher concentrations in tadpoles (*Pérez-Iglesias et al., 2019*). Atrazine causes oxidative stress in several tissues, which can evolve to function loss. The replacement of natural vegetation by agriculture can increase UV incidence and temperature (*Lipinski, Santos & Schuch, 2016*). This is an indirect effect of land use change that can affect the amount of internal melanin in frogs.

Although our statistical model explained much variation in the three substances, other climatic factors that have been changing because of human activities, such as temperature and UV radiation can also change the amount of melanin in internal tissues of frogs (*Franco-Belussi, Provete & De Oliveira, 2017*). Changes in these environmental variables may promote changes in hepatic metabolism, such as increasing glycogen and lipid levels (*Mizell, 1965*; *Barni & Bernocchi, 1991*; *Fenoglio, Bernocchi & Barni, 1992*; *Barni et al., 2002*). Adaptation to natural conditions (i.e., hibernation) may promote an increase in the amount of melanin in the liver by increasing the number of cells (i.e., hyperplasia) or the increase in cell size (i.e., hypertrophy), besides an increase in the production of melanin in cell interior (*Barni et al., 2002*). These changes occur as a mechanism of metabolic defense against free radicals. For example, with increased storage of lipids and glycogen in hepatic tissue due to environmental changes that occur naturally in hibernating animals (*Mizell, 1965*; *Barni & Bernocchi, 1991*; *Fenoglio, Bernocchi & Barni, 1992*; *Barni et al., 2002*). Thus, the liver of anurans is a plastic organ, besides being sensitive to the alterations of the natural biological rhythms (i.e., reproduction and hibernation), coordinating these mechanisms to maintain the homeostasis of the organism during adaptation to environmental changes (*Barni et al., 2002*). Therefore, cells that produce and store melanin seem to be involved in the adaptation to environmental stressors at the tissue level.

*Dendropsophus minutus* and *S. fuscomarginatus* had high amounts, whereas *B. albopunctata* had low amounts of hemosiderin and lipofuscin. Lipofuscin and hemosiderin are products of cellular catabolism and can be used to measure hepatic metabolism, since both substances may be altered as a result of environmental stress following habitat alteration (*Santos et al., 2014*; *Pérez-Iglesias et al., 2016*). Low amounts of these substances indicate decreased phagocytic activity of cells (*Bucke, Vethaak & Lang, 1992*; *Fenoglio et al., 2005*), while accumulation of hemosiderin is related to increased turnover of blood cells (*Fenoglio et al., 2005*). Therefore, the increase in recycling of blood cells in *D. minutus* and *S. fuscomarginatus* indicates that biotic or abiotic factors resulting from anthropic changes may be causing changes in the liver of these two species, but less so in *B. albopunctata*.

The idiosyncratic responses of species may be related to differences in life history traits. Internal melanin varies among species and organs in anurans, possibly due to its adaptive function conferred by the protective functions of the pigment (*Provete et al., 2012*; *Franco-Belussi, Provete & De Oliveira, 2017*). For example, *Physalaemus* and *Leptodactylus* are terrestrial and can have more contact with xenobiotics; while *Boana*, *Scinax*, and

*Dendropsophus* are arboreal and putatively less exposed to aquatic xenobiotics (*Silva et al., 2013*). However, it is noteworthy that any drastic change in land use appears to promote metabolic alterations related to liver physiological processes in anurans. Therefore, MMCs seem to be efficient biomarkers indicating alterations in the liver in response to contrasting land use types. Actions for mitigating the negative effects of industrial agricultural activities must be taken if we want to reach the goal of making environmentally responsible agricultural products (ONU 2050 goals), especially in grassy biomes (*Parr et al., 2014*; *Overbeck et al., 2015*).

## CONCLUSION

Our a priori hypothesis was that frogs from the area with intense agricultural activity would have higher amounts of melanin, hemosiderin, and lipofuscin, because of the increase in hepatic metabolism necessary to deal with potential contaminants and higher solar incidence promoted by loss of vegetation cover. We found that the amount of melanin, hemosiderin, and lipofuscin indeed varied between regions, but each species seemed to respond differently to these contrasting lands use types. The species whose liver metabolism most changed across different land use type were *P. cuvieri*, *B. albopunctata*, and *S. fuscomarginatus*. Therefore, these species should be used for evaluating environmental alterations. Aquatic contaminants may alter organismal health that cannot be assessed by only recording species presence in each environment, since the internal morphology of individuals can be damaged. Alterations in hepatic metabolism can compromise population viability and species persistence in the environment in the long term. Our results reinforce the need to include multiple biological aspects of species (e.g., morphology, physiology) in environmental monitoring programs.

## ACKNOWLEDGEMENTS

B. Valverde, W.R. Rezende, R.A. Assis, and G. Leite helped with histological procedures. R.F. Oliveira kindly prepared the map shown in Fig. 1. M. Almeida-Gomes helped with GIS data processing and extracting area of each land use type. C. ter Braak clarified some statistical aspects of analysis.

### Funding

This study was supported by Conselho Nacional de Desenvolvimento Científico e Tecnológico (CNPq) (grant #477044/2013-1) to Lia Raquel de Souza Santos and Fundação de Amparo a Pesquisa do Estado de São Paulo (FAPESP) (grant #2013/02067-5 and #2018/01078-7) to Classius De Oliveira. This study was financed by the Coordenação de Aperfeiçoamento de Pessoal de Nível Superior, Brasil (CAPES)—Finance Code 001. Lilian Franco-Belussi was supported by a FAPESP post-doctoral fellowship (#2014/00946-4) during the initial phase of this study. Classius De Oliveira has been continuously supported by CNPq #304552/2019-4. Lia Raquel S. Santos received partial funding from the Instituto

Federal Goiano. This work was also supported by Universidade Federal de Mato Grosso do Sul—UFMS/MEC—Brasil. There was no additional external funding received for this study. The funders had no role in study design, data collection and analysis, decision to publish, or preparation of the manuscript.

## Grant Disclosures

The following grant information was disclosed by the authors:
Conselho Nacional de Desenvolvimento Científico e Tecnológico (CNPq): #477044/2013-1.
Fundação de Amparo a Pesquisa do Estado de São Paulo (FAPESP): #2013/02067-5 and #2018/01078-7.
Coordenação de Aperfeiçoamento de Pessoal de Nível Superior, Brasil (CAPES): 001.
FAPESP Post-Doctoral Fellowship: #2014/00946-4.
CNPq: #304552/2019-4.
Universidade Federal de Mato Grosso do Sul—UFMS/MEC.

## Competing Interests

Diogo B. Provete is an Academic Editor for PeerJ.

## Author Contributions

- Lilian Franco-Belussi conceived and designed the experiments, performed the experiments, analyzed the data, prepared figures and/or tables, authored or reviewed drafts of the paper, and approved the final draft.
- Diogo B. Provete analyzed the data, prepared figures and/or tables, authored or reviewed drafts of the paper, and approved the final draft.
- Rinneu E. Borges performed the experiments, authored or reviewed drafts of the paper, and approved the final draft.
- Classius De Oliveira conceived and designed the experiments, authored or reviewed drafts of the paper, and approved the final draft.
- Lia Raquel S. Santos conceived and designed the experiments, performed the experiments, authored or reviewed drafts of the paper, and approved the final draft.

## Animal Ethics

The following information was supplied relating to ethical approvals (i.e., approving body and any reference numbers):

All experimental procedure was approved by the ethics committee on animal use of Universidade Rio Verde (protocol #0316, CEUA/UniRV).

## Field Study Permissions

The following information was supplied relating to field study approvals (i.e., approving body and any reference numbers):

The Instituto Chico Mendes de Conservação da Biodiversidade provided collecting permit (Sisbio #34485-1) and Emas National Park provided housing and authorization for field work.

## Data Availability

The R code used to run all the analyses and associated dataset are publicly available at Mendeley Data: Franco-Belussi, Lilian; Provete, Diogo Borges; Borges, Rinneu; de Oliveira, Classius; Santos, Lia Raquel (2020), "Idiosyncratic liver pigment alterations of five frog species in response to contrasting land use patterns in the Brazilian Cerrado", Mendeley Data, v4 DOI 10.17632/bcsm7v629y.4.

## Supplemental Information

Supplemental information for this article can be found online at http://dx.doi.org/10.7717/peerj.9751#supplemental-information.

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
