# Peer review of "Idiosyncratic liver pigment alterations of five frog species in response to contrasting land use patterns in the Brazilian Cerrado"

_PeerJ, doi:10.7717/peerj.9751_

## Round 0.1 · original submission · Major Revisions

I agree with the assessment of both reviewers, that revisions are needed prior to reconsidering your manuscript for potential publication. Please pay particular attention to the concerns and comments from both reviewers. I also think that the experimental design and methodological approach needs to be better explained, and am concerned about the use of two sites with multiple ponds to draw broad conclusions about the relationship between liver markers and land-use. Please outline in detail how sampling was done from the three ponds at once site and two at the other, for example, how many frogs and of what species came from each? A table would be helpful perhaps with lat longs for each water body. Also please better justify your use of the statistical approach chosen. Like both reviewers I do think this manuscript has merit but needs to be thoroughly revised for further consideration.

Reviewer 1 ·

Basic reporting

In general, the manuscript "Idiosyncratic liver alterations of five frog species to land use changes in the Brazilian Cerrado" is correctly written, in professional English.

I suggest to check the references. There are citations that appear in the text that do not appear in the References section.

Please check text citations: "et al." should be in italic

Figure 1: It is important that the map include a clear scale, to see the distances between the study sites. It would also be useful to observe the location of the ponds on the maps.

Experimental design

The title and objective of the manuscript do not correspond to the experimental design. It is mentioned that the change in land use will be evaluated, but two different scenarios are proposed: a protected site (pristine area) and a site with agricultural activity. If one wants to study changes in land use, one should evaluate the same site before and after a change in land use. These are long-term field studies, which include continuous monitoring. But not having previous information on the studied sites, I suggest to change the way of stating the objective of the work, title and phrases associated with evaluation of changes in land use.
It seems more likely that the study evaluates how melanin, hemosiderin and lipofuscin pigments vary depending on land use.

Materials and Methods Section is correctly written, but some issues must be improved.
Some detailed suggestions:

(Lines 128-130): What are the distance between the ponds in each region? Why did you select three ponds in one region and only two in the other? I suggest you including this information in the text

In lines 139-141, it is not clear how do you structure the campaigns to collect in both sites. Did you collect both sites in the same day, week, month, year? I suggest to clarify this issue, it is important for comparisons.

Its important to know also the conservation status of the selected species (lines 141-144). I suggest you incorporate it.

Lines 157-158: Why did you collect only one sample from only one pond on each region? If you collected samples from each pond.

Lines 161-162: You report results in Materials and Methods section. I suggest to move the data to the Results section.

Validity of the findings

The results are relevant, and very interesting, but, there no do not respond to the stated objective. I suggest restructuring the objective and title of the manuscript.

In addition, there are some results that must be clarified. (Line 241) There is a lack of information about how the occurrence of the species was evaluated. In the Materials and Methods section there is no description of how the monitoring of species was done, or how the occurrence was evaluated.

In line 33, you mention: "Our results add a new perspective to the influence of land use changes on environmental health by highlighting the effect of environmental changes on internal morphological aspects of animals". The experimental design does not allow evaluating the effects of the change in habitat use, but rather the effects observed in frogs that inhabit areas with different land use. Although this results are relevant when evaluating a possible change in land use, and even carrying out environmental risk assessments, it cannot go outside the scope of speculation and great care must be taken when talking about the effects produced by a change in land use.

Additional comments

In general, the manuscript "Idiosyncratic liver alterations of five frog species to land use changes in the Brazilian Cerrado" show relevant results for the ecotoxicology field. The intention of the study is interesting, but many aspects are missing to understand how the research was carried out. And there are some issues regarding the title, objectives and specially Materials and Methods that must be improved.

1- Your most important issue is to check the objectives, title and experimental design and restructure the study so that the experimental design answers the research question being asked
2- More detailed Methology must be introduced. For example, Sampling in just one pond, without knowing the distances between the ponds in the same region, or the dimensions thereof, generates more questions than answers. It is probable that the sample taken is not representative of the region, nor of the pond in question. According to what methodology the water samples, storage, etc. were taken? these are relevant data when validating the results.

Finally, some suggestions are detailed below:

Lines 260-262: please check the wording of that phrase.

Lines 263-279: Could there be a relationship between the amount of liver melanin and exposure to UV radiation due to the lack of refuge sites in the planted pastures? It is interesting if you include this in the Discussion section, taking into account that planted pastures eliminate the Natural refuge sites, such as shrubbery.

Lines 307-310: It would be interesting to give a phylogenetic look that also sheds light on these interspecific differences.

Reviewer 2 ·

Basic reporting

This manuscript examined liver’ melanomacrophages and their metabolic substances including melanin, lipofuscin and hemosiderin to evaluate the land use changes inducing environmental changes. As stated by authors, there are many laboratory studies in this topic but area studies are very important. Contaminants can alter vital functions of internal organs and affect the viability and persistence of species in the environment. To that end liver cell physiology is an important endpoint to determine the effects of land use changes.
Introduction
This part is good organized and clearly emphasize the purpose of the study.
In line 89, …….xenobiotics (…). Here is a literature made by Çakıcı 2015 should also be added to mention melanomacrophage increases. This study revealed an increase in melanomacrophage number of liver tissue after exposure to a pesticide (carbaryl) in an important anuran species, Bufotes variabilis.
(Çakıcı Ö 2015. Histopathologic changes in liver and kidney tissues induced by carbaryl in Bufotes variabilis (Anura: Bufonidae). Experimental and Toxicologic Pathology 67(3), 237–243)
Material and methods
This section is comprehensive and enough to justify the results of study.
Results
There are important findings and these findings are the results of well designed experiment. However, there are no detailed histologic figure. Only one image was found in graphical abstract. And there is no information about which species the liver figure belongs to. Therefore, these histological photographs should be added, if necessary.
Discussion
In this part, results are logical evaluated. And as stated by authors, melanomacroges are important biomarkers showing alterations in the liver in response to land use changes.
Language
English should be checked by a native speaker. In some places, there are grammatical errors.
For example In line 312. Therefore, MMCs seem to be an efficient biomarker that indicated (indicating) alterations in the liver in response to land use change.

Best Regards

Experimental design

This section is comprehensive and enough to justify the results of study.

Validity of the findings

There are important findings and these findings are the results of well designed experiment.

Additional comments

Dear Authors,
This manuscript has valuable findings. As you said, melanomacroges are important biomarkers showing alterations in the liver in response to land use changes. However, I have some suggestions.
-In introduction part: In line 89, …….xenobiotics (....). Here is a literature made by Çakıcı 2015 should also be added to mention melanomacrophage increases. This study revealed an increase in melanomacrophage number of liver tissue after exposure to a pesticide (carbaryl) in an important anuran species, Bufotes variabilis.
(Çakıcı Ö 2015. Histopathologic changes in liver and kidney tissues induced by carbaryl in Bufotes variabilis (Anura: Bufonidae). Experimental and Toxicologic Pathology 67(3), 237–243)
-There are important findings and these findings are the results of well designed experiment. However, there are no detailed histologic figure. Only one image was found in graphical abstract. And there is no information about which species the liver figure belongs to. Therefore, these histological photographs should be added if necessary.
-English should be checked by a native speaker. In some places, there are grammatical errors.
For example In line 312. Therefore, MMCs seem to be an efficient biomarker that indicated (indicating) alterations in the liver in response to land use change.


Best Regards

---

## Round 0.2 · accepted · Accept

We thank the authors for addressing all reviewer concerns. Thank you for better explaining the study's limitations and more clearly linking your findings and aims to the results.

Reviewer 1 ·

Basic reporting

I believe that almost all the points to be improved have been improved, or failing that, the limitations have been explained.

Experimental design

The entire article is much clearer and all my concerns have already been answered.

Validity of the findings

Although the article has some limitations, it is novel and has great potential. The results are relevant in the field of ecotoxicology.

Reviewer 2 ·

Basic reporting

New title "Idiosyncratic liver pigment alterations of five frog species in response to contrasting land use patterns in the Brazilian Cerrado" is more suitable.

Contaminants can alter vital functions of internal organs and affect the viability and persistence of species in the environment. Liver cell physiology is an important endpoint to that reflects important reactions to environmental alterations. This manuscript examined liver’ melanomacrophages and their metabolic substances including melanin, lipofuscin and hemosiderin to evaluate the contrasting land use patterns in five Anuran species.

Experimental design

This section is well designed.

Validity of the findings

Findings are well stated and linked to aim of this study.

Additional comments

In this revision, all corrections were made by authors.